# On the (Non) Injectivity of Piecewise Linear Janossy Pooling

## Abstract

Multiset functions, which are functions that map multisets to vectors, are a fundamental tool in the construction of neural networks for multisets and graphs. To guarantee that the vector representation of the multiset is faithful, it is often desirable to have multiset mappings that are both injective and bi-Lipschitz. Currently, there are several constructions of multiset functions achieving both these guarantees, leading to improved performance in some tasks but often also to higher compute time than standard constructions. Accordingly, it is natural to inquire whether simpler multiset functions achieving the same guarantees are available. In this paper, we make a large step towards giving a negative answer to this question. We consider the family of $k$-ary Janossy pooling, which includes many of the most popular multiset models, and prove that no piecewise linear Janossy pooling function can be injective. On the positive side, we show that when restricted to multisets without multiplicities, even simple deepsets models suffice for injectivity and bi-Lipschitzness. Finally, we provide empirical validation through multiset reconstruction experiments and explicit equal-output constructions for ReLU Janossy maps. The experiments show that reconstruction error increases as point separation decreases, and that the non-injectivity mechanism can be observed directly.

## 1 Introduction

A natural requirement of machine learning models for graphs and point clouds is that they respect the permutation symmetries of the data. A key tool to achieve this is the process of mapping multisets, which are unordered collections of vectors, to a single (ordered) vector which faithfully represents the multiset.

The celebrated deepsets paper (Zaheer et al., 2017) proposed a simple and popular method to map multisets to vectors via elementwise application of a function $f$, followed by sum pooling, namely

$$F(\{\mathbf{x}_1, \ldots, \mathbf{x}_n\}) = \sum_{j=1}^{n} f(\mathbf{x}_j). \tag{1}$$

Another popular alternative, which is more computationally demanding but also more expressive (Zweig & Bruna, 2022), sums a function $f(x_i, x_j)$ over all pairs of points

$$F(\{\mathbf{x}_1, \ldots, \mathbf{x}_n\}) = \sum_{i,j=1}^{n} f(\mathbf{x}_i, \mathbf{x}_j). \tag{2}$$

This type of pairwise summation allows incorporation of relational pooling (Santoro et al., 2017), or attention mechanisms as proposed in the set-transformer paper (Lee et al., 2019).

A natural generalization of both these models is the notion of $k$-ary Janossy pooling, where a function $f$ is applied to all $k$-tuples of the multiset and then summation is applied to all these $k$-tuples. Deepsets models correspond to the case $k = 1$ while set transformers correspond to $k = 2$. Janossy pooling for general $k$ was successfully used in Murphy et al. (2019). *$k$-ary Jaonssy pooling is also related to the concept of $k$-body expansions which is central to the topic of learning inter-atomic potentials (Batatia et al., 2022).*

To ensure the quality of the vector representation of the multiset, a common requirement is that the function $F$ is injective. This requirement enables construction of maximally expressive message passing neural networks (Xu et al., 2018; Morris et al., 2019), and is exploited in a variety of other scenarios where expressivity of graph neural networks is analyzed (Maron et al., 2019; Hordan et al., 2024a; Sverdlov & Dym, 2025; Zhang et al., 2024)

The injectivity requirement can be satisfied even by deepsets models, providing that the function $f : \mathbb{R}^d \to \mathbb{R}^m$ in equation 1 is defined correctly, and the embedding dimension $m$ is large enough. The various aspects of this question are discussed in Wagstaff et al. (2022); Zaheer et al. (2017); Xu et al. (2018); Amir et al. (2023); Tabaghi & Wang (2024); Wang et al. (2024). Most relevant to our discussion are the recent results (Amir et al., 2023) which show that equation 1 can be injective when $f$ is a neural network with smooth activations, but can never be injective when $f$ is a *Continuous Piecewise Linear (CPwL)* function (as is the case when $f$ is a neural network with ReLU activations).

While these theoretical results seem to indicate an advantage of smooth functions $f$ over CPwL ones, empirical evidence indicates that the separation between multisets via smooth activations can be very weak (Bravo et al., 2024; Hordan et al., 2024b), and that empirically the separation obtained even by non-injective CPwL deepsets model is often preferable. Thus, recent papers have argued that a more refined notion of separation is necessary, via the notion of bi-Lipschitz stability (Davidson & Dym, 2025; Amir & Dym, 2025; Balan et al., 2022). In this notion, multisets are required not only to be mapped to distinct vectors, but we also require that the distance between the vector representations resembles the natural Wasserstein distance between the multisets.

In the lens of bi-Lipschitz stability, the ranking of CPwL and smooth multiset functions are reversed. In fact, Amir et al. (2023) and Cahill et al. (2024) showed that smooth multiset functions can never be bi-Lipschitz. In contrast, while CPwL deepsets functions are not injective (or bi-Lipschitz), several recent papers have suggested new CPwL multiset-to-vector mappings based on sorting (Balan et al., 2022; Dym & Gortler, 2024; Balan & Tsoukanis, 2023), Fourier sampling of the quantile function (Amir & Dym, 2025), or max filters (Cahill et al., 2022), and showed that they are both injective and bi-Lipschitz. In fact, Sverdlov et al. (2024) showed that CPwL multiset functions which are injective are automatically also bi-Lipschitz.

Experimentally, it was shown that these CPwL bi-Lipschitz multiset mappings have significant advantages over standard methods, for tasks like learning Wasserstein distances or learning in a low parameter regime (Amir & Dym, 2025) and for graph learning tasks (Davidson & Dym, 2025) including reduction of oversquashing (Sverdlov et al., 2024). On the other hand, these methods are typically more time consuming than standard methods, and at least at the time this paper is written they are not as prevalent as deepsets and set transformers.

The goal of this paper is to address the following question.

**Main Question:** Is it possible to construct CPwL injective (and bi-Lipschitz) functions via $k$-ary pooling?

Currently, we have a negative answer to this question only in the special case where $k = 1$ (deepsets), but for $k \geq 2$ (e.g. set transformers) the answer is unknown. A positive answer to this question would potentially lead to new bi-Lipschitz models which are closer to established models like set transformers, and potentially would have better performance. A negative answer would indicate that bi-Lipschitz models do require different types of multiset functions, such as the sort based functions currently suggested in the literature.

**Main Results**  Our theoretical analysis reveals two main results. The first result is that in full generality, the answer to our main question is negative: $k$-ary Janossy pooling is not injective, except in the unrealistic scenario where $k$ is equal to the full cardinality of the multiset. This result strengthens the argument for using sorting based bi-Lipchitz mappings as in Davidson & Dym (2025); Amir & Dym (2025); Sverdlov et al. (2024).

Our second result shows that the obstruction to injectivity is only the existence of multisets with repeated points: on a compact domain $D$ of multisets where all multisets have distinct points, even the 1-ary CPwL Janossy pooling (deepsets) can be injective and bi-Lipschitz. The computational burden of this construction strongly depends on a constant $R(D)$ which measures how close multisets in $D$ are to have repeated points.

This result suggests that the advantages provided by sort-based methods may only be relevant in datasets where (near) point multiplicity occurs (e.g. point cloud samples of surfaces where points are very close together), and not in datasets where points are typically fairly far away, such as multisets which describe small molecules.

We complement these theoretical findings with controlled empirical experiments. First, in a one-dimensional reconstruction task, we show that ReLU Janossy autoencoders of orders $k = 1, 2, 3$ have substantially larger reconstruction error on low-separation inputs than on well-separated inputs. Second, we explicitly construct distinct multisets whose trained Janossy representations are nearly identical, illustrating the common-affine-region mechanism underlying the proof. We also include a two-dimensional equal-output experiment for random ReLU Janossy maps.

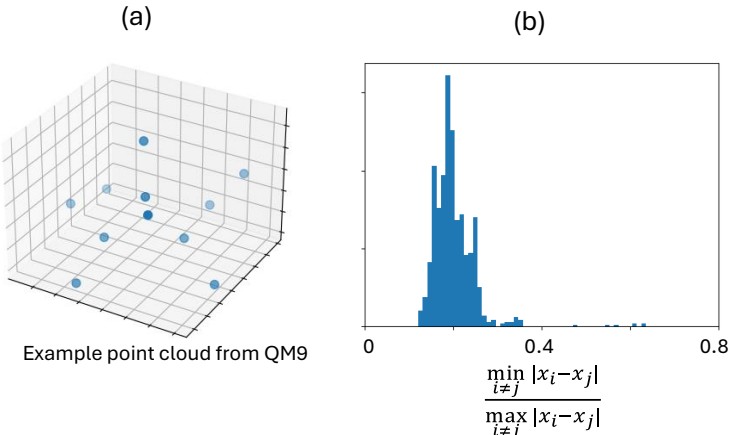

(a) Example point cloud from QM9

(b) $\dfrac{\min\limits_{i \neq j} |x_i - x_j|}{\max\limits_{i \neq j} |x_i - x_j|}$

Figure 1: *This figure illustrates that the assumption that multisets do not have (near)-repeated points is realistic for small molecule datasets. In (a) we show an example multiset from the QM9 (Ruddigkeit et al., 2012; Ramakrishnan et al., 2014) small molecule dataset. This example shows visually that different set elements are not very close together. In (b) we see the statistics of the minimal distance within each multiset (normalized by the maximal distance), over 1,000 representative samples from QM9. In all these instances, the minimal distance was never lower than* 0.1.

**Paper Structure** In Section 2 we will lay out the formal definitions needed to fully define our problem. In Section 3 we state and prove the non-injectivity of CPwL Janossy pooling for general domains, and in Section 4 we state and prove the injectivity results for domains with non-repeated elements. In Section 5 we provide empirical validation of our theoretical results via a multiset reconstruction task. Conclusions, limitations and future work are discussed in Section 6.

## 2 Problem Statement

We begin by formally stating the notions necessary to define our problem. For arbitrary sets $C, Y$, we say that a function $F : C^n \to Y$ is permutation invariant if $F(\mathbf{w}_1, \ldots, \mathbf{w}_n) = F(\mathbf{w}_{\pi(1)}, \ldots, \mathbf{w}_{\pi(n)})$ for every permutation $\pi$ of the coordinates of $\mathbf{w} \in C^n$.

The notion of permutation invariant functions is closely linked to the notion of functions on multisets. A multiset $\{\mathbf{w}_1, \ldots, \mathbf{w}_n\}$ is a collection of elements which is unordered (like sets), but where repetitions are allowed (unlike sets). We denote the space of multisets by $\mathcal{M}_n(C)$.

If $F$ is permutation invariant, we can identify it with a function on multisets in $\mathcal{M}_n(C)$ via

$$F(\{\mathbf{w}_1, \ldots, \mathbf{w}_n\}) = F(\mathbf{w}_1, \ldots, \mathbf{w}_n).$$

Since $F$ is permutation invariant, this expression is well defined and does not depend on the ordering. Conversely, any multiset function $F$ on $\mathcal{M}_n(C)$ can be used to define a permutation invariant function on

$C^n$. Due to this identification, we will use the term 'multiset function' and 'permutation invariant function' alternatingly, according to convenience.

In this paper, our main focus is on permutation invariant functions defined by $k$-ary Janossy pooling. Namely, for some natural numbers $k \leq n$, and a function $f : C^k \to Y$, we define a permutation invariant function $F : C^n \to Y$ via

$$F(\mathbf{x}_1, \ldots, \mathbf{x}_n) = \frac{1}{(n-k)!} \sum_{\pi \in S_n} f\left(\mathbf{x}_{\pi(1)}, \ldots, \mathbf{x}_{\pi(k)}\right). \tag{3}$$

As mentioned above, special cases of Janossy pooling include the deep sets model in equation 1, which corresponds to the case $k = 1$, and set transformer models which correspond to the case $k = 2$ equation 2.

As discussed in the introduction, we will focus on the case where the function $f$ used to define the Janossy pooling is *Continuous Piecewise Linear (CPwL)*. To define this, we recall that a (closed, convex) polytope $P$ is a subset of $\mathbb{R}^d$ defined by a finite number of weak inequalities

$$P = \{\mathbf{x} \in \mathbb{R}^d | \mathbf{a}_j \cdot \mathbf{x} + b_j \geq 0, \forall j = 1, \ldots, J\}.$$

A partition of $\mathbb{R}^d$ is a finite collection of polytopes with non-empty interior, whose union covers $\mathbb{R}^d$ and whose interiors do not intersect. A CPwL function $f : \mathbb{R}^d \to \mathbb{R}^m$ is a continuous function satisfying that, for some partition $\mathcal{P} = \{P_1, \ldots, P_k\}$, the restriction of $f$ to each polytope $P_j$ in the partition is an affine function. The polytopes $P_j$ are called linear regions of $f$. Neural networks defined by piecewise linear activations like ReLU or leaky ReLU are important examples of CPwL functions.

Finally, a multiset function $F : \mathcal{M}_n(C) \to Y$ is *injective* if it is injective in the standard sense: for all distinct multisets $W, W' \in \mathcal{M}_n(C)$ we have $F(W) \neq F(W')$. As discussed in the introduction, the question we discuss in this paper is the injectivity of $F$ induced from Janossy pooling of a CPwL function $f$.

## 3 Non-injectivity of Janossy Pooling for general domains

Now we can state our main theorem:

**Theorem 3.1.** *[Non-Injectivity of $k$-ary Janossy Pooling of CPwL functions] Let $C$ be a subset of $\mathbb{R}^d$ that contains a line segment (usually this will be $[0,1]^d$ or $\mathbb{R}^d$ itself). Let $f : (\mathbb{R}^d)^k \to \mathbb{R}^m$ be a continuous piecewise linear (CPwL) function. Let $n > k$, and let $F : (\mathbb{R}^d)^n \to \mathbb{R}^m$ be the $k$-ary Janossy pooling of $f$. Then $F$ is not injective on $\mathcal{M}_n(C)$.*

To provide intuition for the theorem, we recall the simple proof for the simple case $k = 1, d = 1$, provided in Amir et al. (2023). In this case, we find a pair of distinct points $x, y$ which are in the same linear region of $f$. In this case, the average of $x$ and $y$ is also in the same linear region, and we can use this to obtain a contradiction to injectivity

$$F(x, y) = f(x) + f(y) = f\left(\frac{x+y}{2}\right) + f\left(\frac{x+y}{2}\right) = F\left(\frac{x+y}{2}, \frac{x+y}{2}\right).$$

The proof of the case $k = 1, d = 1$ relies on the trivial observation that we can always find a pair of numbers $(x, y) \in \mathbb{R}^2$ (or more generally in $\mathbb{R}^n$) whose elements are distinct, but come from the same partition which the CPwL function $f : \mathbb{R} \to \mathbb{R}^m$ is subordinate to. To generalize this result to $k$-ary pooling, we will need to show a similar but much stronger property: for any polytope partition of $\mathbb{R}^k$, one can find a vector in $\mathbb{R}^n$ of distinct monotonely decreasing elements, such that all $k$-ary monotonely ordered subvectors belong to a single polytope from the partition:

**Theorem 3.2.** *For every polytope partition $\mathcal{P}$ of $\mathbb{R}^k$, there exists a polytope $P_0 \in \mathcal{P}$ and a point $\mathbf{w} = (w_1, \ldots, w_n) \in (0,1)^n$ such that $w_1 > \cdots > w_n$ and, for any ascending $k$-tuple of indices $i_1 < \cdots < i_k$ in $[n]$, the point $(w_{i_1}, \ldots, w_{i_k})$ is in $\text{int}(P_0)$.*

A visualization of the property described in the theorem is provided in Figure 2 for the special case $k = 2, n = 3$.

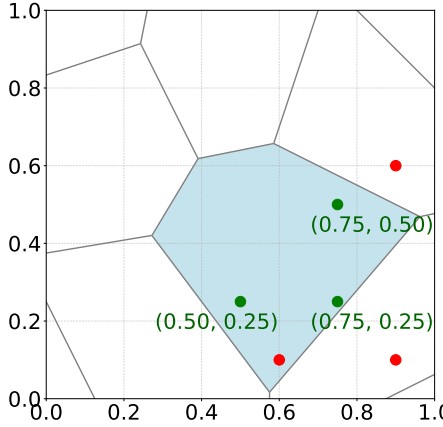

Figure 2: *Visualization of the property from Theorem 3.2 for a polytope partition of $[0,1]^2$. The point $\mathbf{w} = (0.75, 0.5, 0.25)$ fulfills the conditions of the property, as all three 2 dimensional ordered subvectors are in the same linear region (see green dots). The vector $(0.9, 0.6, 0.1)$ does not fulfill the condition (see red dots).*

To the best of our knowledge, Theorem 3.2 has not previously been known and may be of independent interest. The proof of this result is technical and non-trivial and it is given in Appendix A.

We now explain how this proposition can be used to prove Theorem 3.1.

*Proof of Theorem 3.1.* For the sake of simplicity we first prove the theorem in the case $d = 1$. We will later easily generalize this result to the general case $d > 1$. We assume WLOG (without loss of generality) that $C = [0, 1]$.

Let $f : \mathbb{R}^k \to \mathbb{R}^m$ be a CPwL function. Let $F : \mathbb{R}^n \to \mathbb{R}^m$ be its $k$-ary Janossy pooling as in equation 3. Our goal is to prove that $F$ is not injective on multisets $\mathcal{M}_n([0, 1])$.

The first step is to show that $F$ can be defined alternatively by applying Janossy pooling to the permutation invariant function $\hat{f} : \mathbb{R}^k \to \mathbb{R}^m$ defined by

$$\hat{f}(x_1, \ldots, x_k) = \sum_{\pi \in S_k} f\left(x_{\pi(1)}, \ldots, x_{\pi(k)}\right).$$

Note that $\hat{f}$ is a permutation-invariant function and that we can equivalently write

$$F(x_1, \ldots, x_n) = \sum_{1 \leq i_1 < \cdots < i_k \leq n} \hat{f}\left(x_{i_1}, \ldots, x_{i_k}\right),$$

summing over all $\binom{n}{k}$ subsets of $[n]$ of size $k$.

Note that $\hat{f}$ is the sum of finitely many CPwL functions; therefore, it is itself a CPwL function. Let $\mathcal{P}$ be a finite polytope covering of $[0, 1]^k$ such that for all polytopes $P \in \mathcal{P}$, $\hat{f}\big|_P$ is an affine function. Let $P_0$ be as promised from Theorem 3.2, and let $A \in \mathbb{R}^{m \times k}, \mathbf{b} \in \mathbb{R}^m$ such that $\hat{f}\big|_{P_0}(\mathbf{z}) = A\mathbf{z} + \mathbf{b}$. The properties of the point $\mathbf{w}$ in the theorem are preserved under small perturbations. Namely, for some $r > 0$, we have that for all vectors $\boldsymbol{\delta} \in \mathbb{R}^n$ with norm bounded by $r$, we will have both $w_1 + \delta_1 > w_2 + \delta_2 > \ldots > w_n + \delta_n$, and that all $k$ vectors obtained from $\mathbf{w} + \boldsymbol{\delta}$ by choosing $k$ different ascending indices will be in the same polytope $P_0$. It follows that for all such $\boldsymbol{\delta}$

$$\begin{aligned}
F(\mathbf{w} + \boldsymbol{\delta}) &= \sum_{1 \leq i_1 < \cdots < i_k \leq n} \hat{f}\left(w_{i_1} + \delta_{i_1}, \ldots, w_{i_k} + \delta_{i_k}\right) \\
&= \binom{n}{k}\mathbf{b} + \sum_{1 \leq i_1 < \cdots < i_k \leq n} A\left(w_{i_1} + \delta_{i_1}, \ldots, w_{i_k} + \delta_{i_k}\right)^\top \\
&= \binom{n}{k}\mathbf{b} + A\left(\sum_{1 \leq i_1 < \cdots < i_k \leq n} (w_{i_1} + \delta_{i_1}, \ldots, w_{i_k} + \delta_{i_k})^\top\right).
\end{aligned}$$

To contradict injectivity we will want to obtain $F(\mathbf{w}) = F(\mathbf{w} + \boldsymbol{\delta})$, which will hold if

$$\sum_{1 \le i_1 < \cdots < i_k \le n} (\delta_{i_1}, \ldots, \delta_{i_k}) = (0, \ldots, 0).$$

Indeed, this system consists of $k$ linear homogeneous equations in $n > k$ variables. Since there are fewer equations than unknowns, the kernel is at least $(n - k)$-dimensional, so a nonzero solution $\boldsymbol{\delta}$ exists. We can scale this $\boldsymbol{\delta}$ by a sufficiently small number to guarantee that $\|\boldsymbol{\delta}\| < r$. We then have that $F(\mathbf{w}) = F(\mathbf{w} + \boldsymbol{\delta})$, that $\mathbf{w} + \boldsymbol{\delta} \neq \mathbf{w}$, and moreover, since both $\mathbf{w}$ and $\mathbf{w} + \boldsymbol{\delta}$ are sorted from large to small, that $\mathbf{w}$ is not a permutation of $\mathbf{w} + \boldsymbol{\delta}$. Thus $F$ is not injective, and we proved the theorem in the case where $d = 1$.

**When $d > 1$.** We now prove the general case $d > 1$ by a reduction to the case $d = 1$. Let us assume by contradiction that $C \subseteq \mathbb{R}^d$ contains the line segment between some (non-identical) points $\boldsymbol{\alpha}$ and $\boldsymbol{\beta}$, that $f$ is some CPwL function and that the function $F$ obtained by $k$-ary Janossy pooling on $f$ is injective.

Let $g : [0, 1] \to C$ be the affine function $g(t) = (1 - t)\boldsymbol{\alpha} + t\boldsymbol{\beta}$. For any natural $s$, we can extend $g$ to a mapping $g^{(s)} : [0, 1]^s \to (\mathbb{R}^d)^s$ by applying $g$ to each coordinate, i.e., for $\mathbf{t} = (t_1, \ldots, t_s) \in [0, 1]^s$, $g^{(s)}(\mathbf{t}) = (g(t_1), \ldots, g(t_s))$. The function $g^{(s)}$ is affine and injective. Accordingly, the function $F \circ g^{(n)} : [0, 1]^n \to \mathbb{R}^m$ is injective, and we note that it is the Janossy pooling of $f \circ g^{(k)}$ which is a CPwL function as the composition of a CPwL function and an affine function. This leads to a contradiction to our proof for the case $d = 1$. $\square$

### 3.1   Janossy pooling when $k = n$

In the degenerate case where we use $n$-ary pooling for multisets of cardinality $n$, an expensive averaging over all permutations is necessary. In this case we can choose the initial $f$ we use to be a CPwL multiset injective function, such as the sorting based functions constructed in Balan et al. (2022) and mentioned earlier. Since the initial $f$ is already permutation invariant, and $n = k$, we would obtain $F(\mathbf{x}_1, \ldots, \mathbf{x}_n) = n! f(\mathbf{x}_1, \ldots, \mathbf{x}_n)$ in this case, and so Janossy pooling of CPwL functions can be injective in this degenerate case.

## 4   Injectivity under restricted domains

We now discuss our second main result. In contrast to the non-injectivity result presented in the previous section for general multiset domains of the form $\mathcal{M}_n(C)$, we now show that by restricting the domain to a compact $D \subset \mathcal{M}_n(C)$ where each multiset has $n$ distinct elements, injective 1-ary Janossy pooling is possible. To state this theorem formally, we define the natural Wasserstein metric on the space of multiset, and then the notion of a compact set in multiset-space:

**Definition 4.1.** Given two multisets $A, B \in \mathcal{M}_n(\mathbb{R}^d)$, the Wasserstein metric $d_W(A, B)$ is defined as

$$d_W(A, B) = \min_{\sigma \in S_n} \sum_{i=1}^{n} \|\mathbf{a}_i - \mathbf{b}_{\sigma(i)}\|$$

where $A = \{\mathbf{a}_1, \mathbf{a}_2, \ldots, \mathbf{a}_n\}$, $B = \{\mathbf{b}_1, \mathbf{b}_2, \ldots, \mathbf{b}_n\}$, $S_n$ is the set of all permutations of $[n]$, and $\|\cdot\|$ denotes the $\ell_\infty$ norm. Note that the expression above is permutation invariant, and therefore well-defined independently of the order of the elements of $A, B$.

We use the $\ell_\infty$ ground cost because it aligns with the cubical construction below and because $x \mapsto \|x\|_\infty$ is CPwL. By the equivalence of norms, this choice incurs no loss of generality. If $d_{W,2}$ is the analogous Wasserstein distance with Euclidean ground cost, then

$$d_W(A, B) \le d_{W,2}(A, B) \le \sqrt{d}\, d_W(A, B).$$

Consequently, using $\ell_\infty$ preserves compactness, injectivity, and bi-Lipschitz stability, affecting only the specific Lipschitz constants.

**Definition 4.2.** Let $C \subset \mathbb{R}^d$. We say that $D \subset \mathcal{M}_n(C)$ is compact if every sequence of multisets $\{A_j\}_{j=1}^{\infty}$ in $D$ has a subsequence that converges to a multiset in $D$ with respect to the Wasserstein metric $d_W$. This is the standard definition of compactness in a metric space.

We can now state our second main result: in the absence of multisets with repeated elements, even 1-ary pooling is injective:

**Theorem 4.3.** *Let $D \subset \mathcal{M}_n(C)$ be a compact set of multisets where each multiset has $n$ distinct elements. Then there exists some $m = m(D)$ and a continuous piecewise linear function $f : \mathbb{R}^d \to \mathbb{R}^m$ such that its 1-ary Janossy pooling $F(A) = \sum_{\mathbf{a} \in A} f(\mathbf{a})$ is injective on $D$, and bi-Lipschitz with respect to the Wasserstein distance.*

We will prove this theorem by construction. We begin with some preliminaries: we first introduce the minimal separation function $r : D \to \mathbb{R}_{\geq 0}$ via

$$r(A) = \min_{\substack{\mathbf{a}_i, \mathbf{a}_j \in A \\ i \neq j}} \|\mathbf{a}_i - \mathbf{a}_j\|$$

for any multiset $A = \{\mathbf{a}_1, \mathbf{a}_2, ..., \mathbf{a}_n\} \in D$, where $\|\cdot\|$ denotes the $\ell_\infty$ norm. We note that by our theorem's assumptions, $r(A) > 0$ for all $A \in D$. We next define $R(D)$ to be the minimal separation obtained on all of $D$, namely

$$R(D) = \inf_{A \in D} r(A) = \inf_{A \in D} \min_{\substack{\mathbf{a}_i, \mathbf{a}_j \in A \\ i \neq j}} \|\mathbf{a}_i - \mathbf{a}_j\|.$$

We next show that, due to the compactness of $D$, the infimum in the definition of $R(D)$ is obtained and $R(D)$ is always strictly positive.

**Proposition 4.4.** *If $D \subset \mathcal{M}_n(C)$ is a compact set of multisets, where each multiset $A \in D$ consists of $n$ distinct elements, then its minimum separation $R(D)$ is positive.*

*Proof.* Assume, for the sake of contradiction, that $R(D) = 0$. By the definition of the infimum, this implies that there exists a sequence of multisets $\{A_j\}_{j=1}^\infty$ in $D$ such that $r(A_j) \to 0$ as $j \to \infty$. Each $A_j = \{\mathbf{a}_1^{(j)}, \ldots, \mathbf{a}_n^{(j)}\}$ consists of $n$ distinct points.

Since $D$ is compact, the sequence $\{A_j\}_{j=1}^\infty$ has a subsequence $\{A_{j_l}\}_{l=1}^\infty$ that converges to a multiset $A^* \in D$ with respect to the Wasserstein metric $d_W$. Let $A^* = \{\mathbf{a}_1^*, \ldots, \mathbf{a}_n^*\}$. By definition of $D$, we have $r(A^*) > 0$.

Let $\epsilon = \frac{r(A^*)}{2}$. From the convergence of $r(A_{j_l})$ and $A_{j_l}$, there exists an $l$ such that $r(A_{j_l}) < \epsilon$ and $d_W(A_{j_l}, A^*) < \epsilon$. For this $l$, we deduce there are at least two distinct points, WLOG $a_1^{(j_l)}, a_2^{(j_l)} \in A_{j_l}$, such that $\|a_1^{(j_l)} - a_2^{(j_l)}\| < \epsilon$. Next, let $\sigma \in S_n$ be the permutation such that the minimum in the definition of the Wasserstein distance between $A^*, A_{j_l}$ is attained. Then, applying the triangle inequality twice, we get:

$$\frac{r(A^*)}{2} = \epsilon > d_W(A_{j_l}, A^*)$$
$$= \sum_{i=1}^n \|\mathbf{a}_i^{(j_l)} - \mathbf{a}_{\sigma(i)}^*\|$$
$$\geq \|\mathbf{a}_1^{(j_l)} - \mathbf{a}_{\sigma(1)}^*\| + \|\mathbf{a}_2^{(j_l)} - \mathbf{a}_{\sigma(2)}^*\|$$
$$\geq \|\mathbf{a}_{\sigma(1)}^* - \mathbf{a}_{\sigma(2)}^* + \mathbf{a}_2^{(j_l)} - \mathbf{a}_1^{(j_l)}\|$$
$$\geq \|\mathbf{a}_{\sigma(1)}^* - \mathbf{a}_{\sigma(2)}^*\| - \|\mathbf{a}_1^{(j_l)} - \mathbf{a}_2^{(j_l)}\|$$
$$\geq r(A^*) - \epsilon = \frac{r(A^*)}{2}.$$

This is a contradiction. We conclude that $R(D) > 0$. $\square$

We now provide the construction of the function $f$. Tessellate $\mathbb{R}^d$ with a grid of non-overlapping, adjacent $d$-dimensional hypercubes $Q_k$, each with side length $s = \frac{R(D)}{2}$.

We define a $\delta$-margin around each hypercube $Q_k$ using the $\ell_\infty$ distance. For any point $\mathbf{x}$, its $\ell_\infty$ distance to the hypercube $Q_k$ is given by $d_\infty(\mathbf{x}, Q_k) = \min_{\mathbf{y} \in Q_k} \|\mathbf{x} - \mathbf{y}\|_\infty$. The $\delta$-margin of $Q_k$ is then the set of points $\{\mathbf{x} \in \mathbb{R}^d \setminus Q_k \mid d_\infty(\mathbf{x}, Q_k) < \delta\}$.

Let $\delta$ be a margin width chosen such that $0 < \delta < \frac{R(D)}{4}$. This ensures that the equation $(s + 2\delta) < R(D)$ is satisfied. This implies that if a hypercube $Q_k$ together with its $\delta$-margin contains a point $\mathbf{a} \in A$ (for $A \in D$), it cannot contain any other point $\mathbf{a}' \in A \setminus \{\mathbf{a}\}$. In particular, $Q_k$ itself, can contain at most one point from $A$.

Let $\mathcal{I}$ be the finite set of indices of hypercubes $Q_k$ that intersect $C' = \bigcup_{A \in D} A \subseteq C$. Since $D$ is compact, $C'$ is bounded, ensuring $\mathcal{I}$ is finite.

For each hypercube $Q \in \{Q_k\}_{k \in \mathcal{I}}$, we define a local $(d + 1)$-dimensional feature vector $f_Q(\mathbf{x})$, consisting of two components:

**Indicator Component $f_{Q,\mathbf{ind}}(\mathbf{x}) \in [0, 1]$:** $f_{Q,\text{ind}}(\mathbf{x}) = 1$ if $\mathbf{x} \in Q$, and $f_{Q,\text{ind}}(\mathbf{x}) = \max(0, 1 - d_\infty(\mathbf{x}, Q)/\delta)$ elsewhere. This ensures $f_{Q,\text{ind}}(\mathbf{x}) = 1 \iff \mathbf{x} \in Q$, and that the support of $f_{Q,\text{ind}}$ is precisely $Q$ together with its $\delta$-margin. Note that this component is CPwL.

**Relative Coordinate Component $(\mathbf{f}_{Q,\mathbf{coords}}(\mathbf{x}) \in \mathbb{R}^d)$:** This is a CPwL function defined by the following properties:

- If $\mathbf{x} \in Q$, then $\mathbf{f}_{Q,\text{coords}}(\mathbf{x}) = \mathbf{x}$.

- If $\mathbf{x}$ is located outside $Q$ and its $\delta$-margin (i.e., $d_\infty(\mathbf{x}, Q) \geq \delta$), then $\mathbf{f}_{Q,\text{coords}}(\mathbf{x}) = \mathbf{0}$.

- In the $\delta$-margin (i.e., for $\mathbf{x}$ such that $0 < d_\infty(\mathbf{x}, Q) < \delta$), $\mathbf{f}_{Q,\text{coords}}(\mathbf{x})$ interpolates continuously and piecewise linearly between the values at $\partial Q$, and $\mathbf{0}$ at the outer boundary of the margin.

We shall now demonstrate how a function satisfying the third condition can be constructed. By Goodman & Pach (1988), the $\delta$-margin can be triangulated without introducing new vertices such that each simplex of the triangulation contains vertices belonging to both $Q$ and the outer border of the margin.

It is well known that given a simplex in $\mathbb{R}^d$ defined by $d + 1$ affinely independent points $p_0, \ldots, p_d$, and corresponding values $y_0, \ldots, y_d$, there exists a unique affine function $h$ such that $h(x_i) = y_i$ for all $i$.

Applying this to our triangulated $\delta$-margin, we define $f$ piecewise over each simplex by assigning the known values of $f$ at its vertices—values from $Q$ and zeros from the outer border. The unique affine interpolation over each simplex ensures that $f$ transitions continuously between the identity on $Q$ and zero on the outer region, satisfying the desired conditions.

Constructions of this sort are standard in numerical analysis and finite element methods. For instance, in the context of simplicial finite elements, the $\mathbb{P}_1$ interpolant of a function $v$ is the unique piecewise affine function that coincides with $v$ at the mesh vertices (see, e.g., (Brenner & Scott, 2008, 3.3)).

The function $f_Q(\mathbf{x}) = (f_{Q,\text{ind}}(\mathbf{x}), \mathbf{f}_{Q,\text{coords}}(\mathbf{x}))$ is therefore CPwL.

The overall function $f : \mathbb{R}^d \to \mathbb{R}^m$ is the concatenation $f(\mathbf{x}) = (\ldots, f_{Q_k}(\mathbf{x}), \ldots)_{k \in \mathcal{I}}$. The output dimension is $m = |\mathcal{I}| \cdot (d + 1)$.

Now that we have defined the CPwL function $f$ we will use for the proof, we formally conclude the proof:

*Proof of Theorem 4.3.* Let $A \in D$ be a multiset $\{\mathbf{a}_1, \ldots, \mathbf{a}_n\}$. The 1-ary Janossy pooling is $F(A) = \sum_{j=1}^n f(\mathbf{a}_j)$. We show $A$ can be uniquely recovered from $F(A)$. Let $F_{Q_k,\text{ind}}$ and $\mathbf{F}_{Q_k,\text{coords}}$ be the components of $F(A)$ corresponding to $Q_k$. We first prove two simple lemmas

**Lemma 4.5.** *$F_{Q_k,ind}(A) = 1$ if and only if there exists a unique $\mathbf{a} \in A$ such that $\mathbf{a} \in Q_k$ and no element of $A$ lies in the $\delta$-margin of $Q_k$.*

*Proof.* $(\Rightarrow)$ Suppose $F_{Q_k,\text{ind}}(A) = 1$. Assume for the sake of contradiction that there is no $\mathbf{a} \in A$ such that $\mathbf{a} \in Q_k$. The support of $f_{Q,\text{ind}}$ is $Q_k$ together with its $\delta$-margin. For all $\mathbf{a}$ in this margin, $0 < f_{Q,\text{ind}}(\mathbf{a}) < 1$. However,

$$F_{Q_k,\text{ind}}(A) = \sum_{j=1}^n f_{Q_k,\text{ind}}(\mathbf{a}_j) = 1.$$

This implies that at least two elements of $A$ lie in the $\delta$-margin of $Q_k$, which is a contradiction to the separation condition that $\delta$ was constructed to satisfy.

($\Leftarrow$) Suppose there exists $\mathbf{a} \in A$ such that $\mathbf{a} \in Q_k$. By construction of $s$ and $\delta$, no other element of $A$ lies in the support of $f_{Q,\text{ind}}$. Therefore,

$$F_{Q_k,\text{ind}}(A) = f_{Q_k,\text{ind}}(\mathbf{a}) = 1.$$

$\square$

**Lemma 4.6.** *If $\mathbf{a} \in A$ lies in a hypercube $Q_k$, then*

$$\mathbf{F}_{Q_k,coords}(A) = \mathbf{f}_{Q_k,coords}(\mathbf{a}) = \mathbf{a}.$$

*Proof.* The proof is similar to the direction ($\Leftarrow$) in the proof of 4.5. $\square$

We now prove that $A$ can be recovered uniquely from $F(A)$. We do this using the following procedure. We go over all hypercubes $Q_k$. We then check whether $F_{Q_k,\text{ind}}(A) = 1$. By Lemma 4.5 we know that this is the case if and only if $A$ contained an element in $Q_k$, and in this case the element is unique. We can now recover this element from $F_{Q_k,\text{coords}}$ using Lemma 4.6. We have thus uniquely recovered all elements of $A$. We note that if $A$ contains elements which are in the intersection of several hypercubes, this reconstruction procedure will give us the same elements of $A$ from several different hypercubes. This does not cause any issues since we know that $A$ does not contain multiplicities.

Finally, to prove the bi-Lipschitzness of the construction: we note that the set $D$ could be covered by a finite union of polytopes (e.g. hypercubes) so that the union of all these hypercubes $\hat{D}$ contains $D$ but still does not contain multisets with repeated elements. As we now proved, we can construct a CPwL function $f$ so that the resulting $F$ obtained from 1-ary Janossy pooling will be injective on $\hat{D}$. Since $F$ and the 1-Wasserstein distance are both CPwL functions which attain the same zeros on $\hat{D} \times \hat{D}$, and $\hat{D} \times \hat{D}$ can be written as a finite union of compact polytopes, we can apply (Sverdlov et al., 2024, Lemma 3.4) to show that $F$ is bi-Lipschitz on each polytope separately, and therefore also on the union which gives us $\hat{D} \times \hat{D}$. $\square$

## 4.1 Dependence on separation

We note that the dimension $m$ which $F, f$ map to in the construction, depends strongly on the separation $R(D)$ and the dimension $d$. If we add the assumption that all elements of multisets $A$ are in the unit cube $[0,1]^d$, then a tessellation of side length $\sim R(D)/2$ would require an embedding dimension of $m \sim (d+1) \cdot (2/R(D))^d$. This suggests that when $D$ contains elements which are 'almost identical', so that $R(D)$ is small, then 1-ary pooling may not really be enough to get a good embedding with an affordable function $f$. We emphasize that this is the cost of our particular tessellation-based construction, not a lower bound on the dimension needed for injective CPwL pooling. More efficient constructions may be possible. Still, any minimal-separation based construction should be expected to depend on the dimension $d$ and the resolution $R(D)^{-1}$.

As shortly discussed in the introduction, one possible example where the separation $R(D)$ is reasonably large is small molecules. To examine this, we randomly chose 1000 molecules from the QM9 (Ruddigkeit et al., 2012; Ramakrishnan et al., 2014) small molecule datasets. Each molecule is represented as a multiset of vectors residing in $\mathbb{R}^3$. For each of these multisets, we computed the minimum distance between multiset elements, and normalized it by the maximal distance between elements. A histogram of the results is shown in Figure 1(b). We see that in all instances the ratio was never lower than 0.1, so we can estimate that a ratio of $R(D) \approx 0.1$ could be reasonable for this type of problem.

For this QM9 setting, $d = 3$ and $R(D) \approx 0.1$ give roughly $(2/0.1)^3$ cubes and about $(d+1)20^3 \approx 32000$ output coordinates in our construction. This scale is not prohibitive in modern machine learning, but the dependence on $R(D)^{-d}$ can become costly in higher ambient dimension or for substantially smaller separation.

## 5 Empirical Validation

Section 3 establishes that CPwL Janossy pooling is non-injective on general multiset domains, while Section 4 shows that injectivity and bi-Lipschitzness are achievable on domains with a positive minimum point separation $R(D) > 0$. We now give controlled experiments illustrating both aspects.

First, we test whether trained Janossy autoencoders reconstruct separated multisets more accurately than nearly degenerate multisets. Second, we directly construct pairs of distinct multisets whose trained CPwL Janossy encodings nearly coincide.

## 5.1 Reconstruction versus separation

We consider one-dimensional multisets

$$X = \{x_1, \ldots, x_{10}\} \in \mathcal{M}_{10}([0,1]).$$

For each multiset, we measure its minimum separation by

$$r(X) = \min_{i \neq j} |x_i - x_j|,$$

which is the one-dimensional version of the separation quantity used in Section 4. We train $k$-ary Janossy autoencoders for $k \in \{1, 2, 3\}$. The encoder has the form

$$E_{k,\theta}(X) = \frac{1}{k!\binom{10}{k}} \sum_{1 \leq i_1 < \cdots < i_k \leq 10} \sum_{\pi \in S_k} f_\theta\big(x_{i_{\pi(1)}}, \ldots, x_{i_{\pi(k)}}\big),$$

where $f_\theta$ is a ReLU MLP. (A matched $k = 2$ GELU comparison is reported in Appendix C.1.) The decoder maps the latent vector back to ten scalar outputs. Since the inputs are one-dimensional, we train the decoder to output the sorted input multiset. Namely, if

$$D_\phi(E_{k,\theta}(X)) = (\hat{x}_1, \ldots, \hat{x}_{10}),$$

then the training loss is

$$\mathcal{L}(X) = \frac{1}{10} \sum_{i=1}^{10} \big(\hat{x}_i - x_{(i)}\big)^2,$$

where $x_{(1)} \leq \cdots \leq x_{(10)}$ denotes the sorted input.

For each $k \in \{1, 2, 3\}$, the Janossy feature map $f_\theta : \mathbb{R}^k \to \mathbb{R}^{16}$ is a one-hidden-layer ReLU MLP of width 64. The decoder $D_\phi : \mathbb{R}^{16} \to \mathbb{R}^{10}$ is a two-hidden-layer ReLU MLP of width 64. All models are trained with Adam, learning rate $3.5 \cdot 10^{-3}$, weight decay $10^{-6}$, batch size 512, and 450 epochs.

Training samples are drawn from a mixture distribution. Half are i.i.d. uniform multisets in $[0, 1]$. For the other half, we sample $\rho \sim \text{Unif}(0, 0.09)$, draw $u_1, \ldots, u_{10}$ uniformly in $[0, 1 - 9\rho]$, sort them as $u_{(1)} \leq \cdots \leq u_{(10)}$, and set

$$x_{(i)} = u_{(i)} + (i - 1)\rho.$$

This produces controlled-separation samples with $r(X) \geq \rho$. Evaluation uses 25,000 controlled-separation samples. Samples are then binned by their actual minimum separation $r(X)$. The shaded bands in Figure 3 denote one standard error of the mean within each bin.

Table 1 and figure 3 show a consistent separation-dependent degradation. For all three Janossy orders, low-separation inputs have roughly $8\times$ larger reconstruction error than high-separation inputs. Increasing $k$ does not remove the separation dependence.

## 5.2 Explicit encoder near-collisions

To more directly demonstrate the non-injectivity mechanism, we construct pairs of distinct multisets whose trained ReLU Janossy encodings nearly coincide, using the k=2 autoencoder we trained in 5.1.

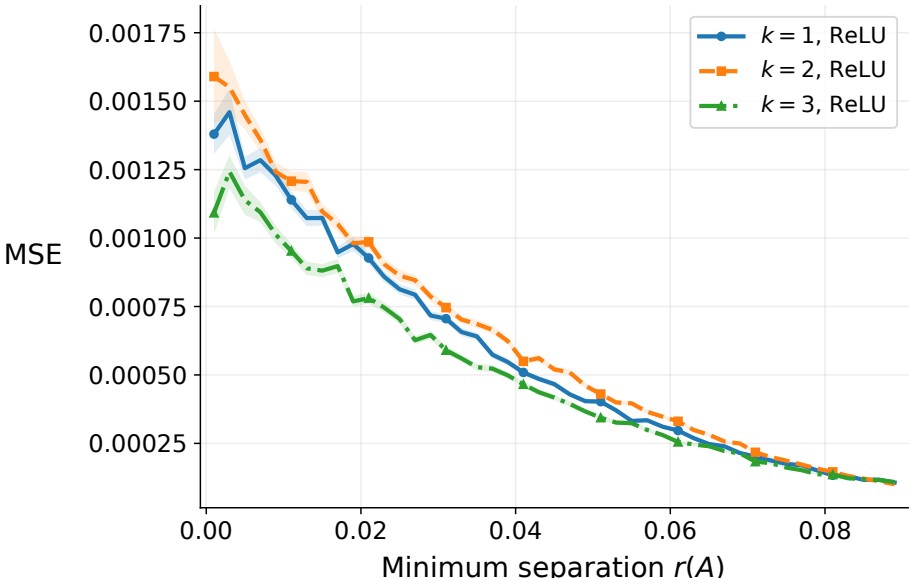

Figure 3: Reconstruction error versus minimum separation in one dimension. Each curve reports binned sorted-MSE reconstruction error for a ReLU $k$-ary Janossy autoencoder. The shaded region is one standard error of the mean within each separation bin. For all fixed orders tested, reconstruction becomes substantially worse as the minimum separation approaches zero.

| Model | Low-sep. MSE | High-sep. MSE | Ratio |
|---|---|---|---|
| $k = 1$, ReLU | $1.285 \cdot 10^{-3}$ | $1.492 \cdot 10^{-4}$ | 8.62 |
| $k = 2$, ReLU | $1.372 \cdot 10^{-3}$ | $1.545 \cdot 10^{-4}$ | 8.88 |
| $k = 3$, ReLU | $1.097 \cdot 10^{-3}$ | $1.410 \cdot 10^{-4}$ | 7.78 |

Table 1: Low- and high-separation reconstruction error for one-dimensional ReLU Janossy autoencoders. The low-separation bin is $r(X) \in [0, 0.01]$, and the high-separation bin is $r(X) \in [0.07, 0.09]$.

For $k = 2$, the proof of Theorem 3.1 suggests perturbing a sorted multiset $X = (x_1, \ldots, x_n)$ by a vector $\mathbf{h}$ satisfying the linear constraints

$$\sum_{1 \leq i < j \leq n} (h_i, h_j) = \mathbf{0}.$$

If all tuple inputs remain in the same affine region of the ReLU MLP, such perturbations leave the Janossy sum unchanged. We use this construction to generate 'clustered' examples, and we also test 'relaxed' examples where the same kind of nullspace perturbations are applied without the strict 'same affine region' condition.

Figure 4 shows two representative examples. The first is a clustered, proof-style near-collision. The second is a relaxed, more visually separated near-collision. In both cases, $X \neq X'$, and the encoder outputs and decoded multisets are much closer than the inputs.

Appendix C.2 gives a two-dimensional experiment showing the non-injectivity mechanism. For four random ReLU $k = 2$ Janossy maps, we construct two-dimensional point clouds $X, X' \subset [0, 1]^2$ with positive Wasserstein distance whose outputs coincide up to floating-point precision.

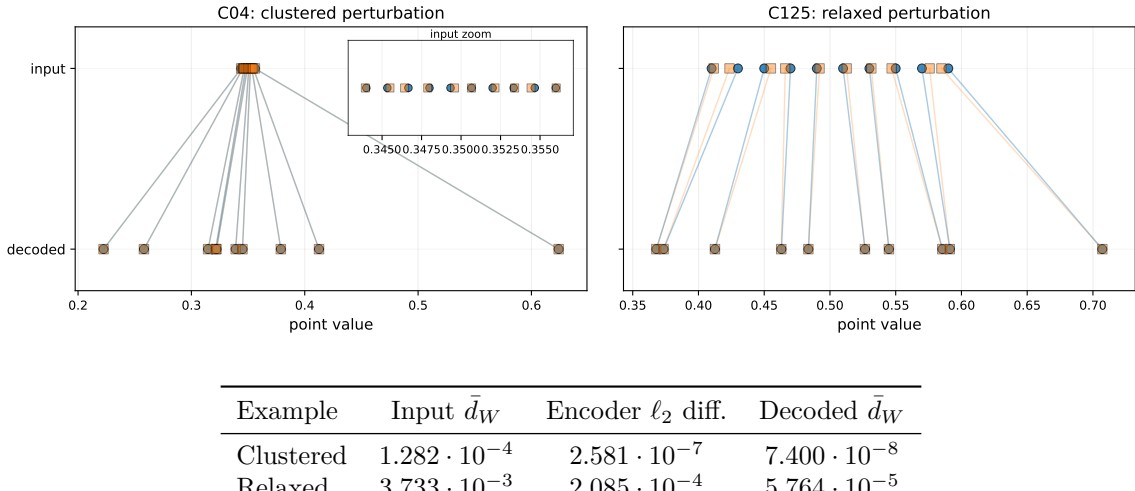

| Example | Input $\bar{d}_W$ | Encoder $\ell_2$ diff. | Decoded $\bar{d}_W$ |
|---|---|---|---|
| Clustered | $1.282 \cdot 10^{-4}$ | $2.581 \cdot 10^{-7}$ | $7.400 \cdot 10^{-8}$ |
| Relaxed | $3.733 \cdot 10^{-3}$ | $2.085 \cdot 10^{-4}$ | $5.764 \cdot 10^{-5}$ |

Figure 4: Direct near-collisions for the trained ReLU $k = 2$ Janossy encoder. **Top:** Each panel shows two input multisets $X, X'$ on the upper row and the corresponding decoded multisets $D(E(X)), D(E(X'))$ on the lower row. Blue circles correspond to $X$ and $D(E(X))$, while transparent orange squares correspond to $X'$ and $D(E(X'))$. Lines connect corresponding sorted input and decoded points. Although the input multisets are distinct, the encoded and decoded outputs nearly coincide. **Bottom:** Normalized Wasserstein distance $\bar{d}_W = d_W/n$ for the input and decoded multisets corresponding to the plotted examples.

## 6 Conclusion, limitations and future work

In this paper, we established two main results regarding continuous piecewise linear (CPwL) Janossy pooling: (a) it is not injective on general multiset domains when $k < n$, and (b) on compact domains of multisets with non-repeated points, even 1-ary CPwL Janossy pooling can be injective and bi-Lipschitz. Our empirical study supports this picture: reconstruction error increases on low-separation inputs, and explicit equal-output constructions exhibit the common-affine-region mechanism underlying the non-injectivity result.

These conclusions should not be read as predicting that architectures such as Deep Sets (Zaheer et al., 2017) or Set Transformers (Lee et al., 2019) should fail in practice. These models often perform well in downstream tasks, and theoretically are known to approximate all continuous multiset functions (Zaheer et al., 2017; Murphy et al., 2019; Wagstaff et al., 2022). Moreover, versions of these models which use analytic activations rather than CPwL can achieve injectivity guarantees (Amir et al., 2023). However, such constructions will not be bi-Lipschitz Amir et al. (2023); Davidson & Dym (2025).

Bi-Lipschitz stability was shown to be useful for downstream tasks which require prediction of invariant metrics (Shivottam & Mishra, 2026) or in a low parameter regime (Amir & Dym, 2025; Davidson & Dym, 2025). Moreover, CPwL and homogenous multiset mappings which are injective are automatically bi-Lipschitz (Sverdlov et al., 2024), and it seems that all known bi-Lipschitz constructions are CPwL. This motivated our focus on $k$-ary pooling using CPwL functions: if we could construct injective multiset functions via $k$-ary pooling over CPwL functions, then we would automatically obtain the bi-Lipschitzness. Unfortunately, our results show that for general domains, such injectivity is not possible, thus justifying the use of previously suggested injective CPwL multiset embeddings, such as sorting based, FSW, or max-filtering (Balan et al., 2022; Amir & Dym, 2025; Cahill et al., 2022). On the positive side, on domains which do not contain near-repeated points, simple CPwL deepsets models suffice for injectivity and bi-Lipschitz guarantees.

Building upon our positive result for 1-ary CPwL Janossy pooling on domains of sets (i.e., multisets with point multiplicities of at most one), a natural direction for future work is to explore the capacity of higher-order pooling. We conjecture that for a given integer $k \geq 1$, $k$-ary CPwL Janossy pooling can be injective on compact domains of multisets where the multiplicity of any individual element is at most $k$. In Appendix B we formulate this question more precisely using the distance to a $(k + 1)$-fold multiplicity.

A limitation of this work is that we only analyze the injectivity of CPwL Janossy pooling. There are many functions which are neither CPwL nor smooth. An interesting avenue for future work is investigating whether such functions can be used to construct injective and bi-Lipschitz multiset functions via $k$-ary pooling, and whether these can lead to multiset models with good empirical performance. This question is most interesting for $k = 2$ as 2-ary Janossy pooling has reasonable complexity, and as for $k = 1$ such a function can only exist if it is not differentiable at any point (Amir et al., 2023).

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

# A   Technical Appendix

## A.1   Proof of Theorem 3.2

We first establish the following supporting result:

Let $\mathbf{v} \in [0,1]^k$. We define $\text{POLY}(\mathbf{v}) = \{P \in \mathcal{P} \mid \mathbf{v} \in P\}$ to be the set of polytopes in the covering $\mathcal{P}$ that contain the point $\mathbf{v}$.

**Lemma A.1.** *Let $\mathbf{v} \in \mathbb{R}^k$. Let $\mathcal{P}$ be a finite polytope covering of $\mathbb{R}^k$. There exists $\epsilon > 0$ such that the $\epsilon$-ball around $\mathbf{v}$ w.r.t. the $\ell_1$ metric, denoted by $B_{\ell_1}(\mathbf{v}, \epsilon) = \{\mathbf{x} \in \mathbb{R}^k : \|\mathbf{x} - \mathbf{v}\|_1 < \epsilon\}$, does not intersect any polytope that does not contain $\mathbf{v}$:*

$$B_{\ell_1}(\mathbf{v}, \epsilon) \cap \bigcup (\mathcal{P} \setminus \text{POLY}(\mathbf{v})) = \emptyset.$$

*Proof of Lemma A.1.* Let $P \in \mathcal{P} \setminus POLY(\mathbf{v})$. Since $P$ is closed,

$$\text{dist}(\mathbf{v}, P) = \inf_{\mathbf{x} \in P} \|\mathbf{x} - \mathbf{v}\| > 0.$$

Let

$$\epsilon = \frac{1}{2} \min_{P \in \mathcal{P} \setminus \text{POLY}(\mathbf{v})} (\text{dist}(\mathbf{v}, P)).$$

Since $\mathcal{P}$ is finite, $\epsilon$ is well defined and positive. For this choice of $\epsilon$, we have

$$B_{\ell_1}(\mathbf{v}, \epsilon) \cap \bigcup (\mathcal{P} \setminus \text{POLY}(\mathbf{v})) = \emptyset.$$

$\square$

*Proof of Theorem 3.2.* Fix some $x \in (0,1)$ and let $\mathbf{v}_0 = (x, \ldots, x) \in (0,1)^k$.

Using Lemma A.1, let $\epsilon_1 > 0$ such that $B_{\ell_1}(\mathbf{v}_0, \epsilon_1) \subset \bigcup \text{POLY}(\mathbf{v}_0)$ and $B_{\ell_1}(\mathbf{v}_0, \epsilon_1) \subset (0,1)^k$.

Let $\mathbf{v}_1 = \mathbf{v}_0 + \frac{\epsilon_1}{2}\mathbf{e}_1 = (x + \frac{\epsilon_1}{2}, x, \ldots, x)$.

We continue this construction by an inductive process. For all $1 < i \leq k$:

Let $\epsilon_i > 0$ such that $B_{\ell_1}(\mathbf{v}_{i-1}, \epsilon_i) \subset \bigcup \text{POLY}(\mathbf{v}_{i-1})$ and $\epsilon_i < \frac{\epsilon_{i-1}}{2}$.

Let $\mathbf{v}_i = \mathbf{v}_{i-1} + \mathbf{e}_i \frac{\epsilon_i}{2} = (x + \frac{\epsilon_1}{2}, \ldots, x + \frac{\epsilon_i}{2}, x, \ldots, x)$.

In the end of this process we get a sequence of $k + 1$ vectors $\mathbf{v}_0, \ldots, \mathbf{v}_k$ which are all in $\mathbb{R}^k_{\text{sorted}} := \{\mathbf{y} \in \mathbb{R}^k \mid y_1 \geq y_2 \geq \ldots \geq y_k\}$.

**Proposition A.2.** $\mathrm{POLY}(\mathbf{v}_k) \subset \cdots \subset \mathrm{POLY}(\mathbf{v}_0)$.

*Proof.* Let $i \in [k]$. Note that $\mathbf{v}_i - \mathbf{v}_{i-1} = \frac{\epsilon_i}{2}\mathbf{e}_i \Rightarrow \mathbf{v}_i \in B_{\ell_1}(\mathbf{v}_{i-1}, \epsilon_i)$. Assume, for the sake of contradiction, that there exists $P \notin \mathrm{POLY}(\mathbf{v}_{i-1})$ such that $\mathbf{v}_i \in P$. Then

$$\mathbf{v}_i \in B_{\ell_1}(\mathbf{v}_{i-1}, \epsilon_i) \cap \bigcup (\mathcal{P} \setminus \mathrm{POLY}(\mathbf{v}_{i-1})) = \emptyset,$$

which is a contradiction. $\square$

**Proposition A.3.** *There exists a single polytope $P_0 \in \mathcal{P}$ such that $\mathbf{v}_k \in P_0$; in particular, $\mathbf{v}_k$ lies in the interior of this polytope.*

*Proof.* Assume, for the sake of contradiction, that $|\mathrm{POLY}(\mathbf{v}_k)| > 1$. Let $P_0, P_1 \in \mathrm{POLY}(\mathbf{v}_k)$ be two different polytopes. Convexity is preserved under intersection, therefore $P_0 \cap P_1$ is a convex set. Under the assumption that the interiors of polytopes in $\mathcal{P}$ do not intersect, we see that $P_0 \cap P_1$ has an empty interior; therefore, there exists a hyperplane $H \subset \mathbb{R}^k$ such that $P_0 \cap P_1 \subset H$ (see (Boyd & Vandenberghe, 2004, 2.5.2)).

By Proposition A.2, we have $\mathbf{v}_0, \ldots, \mathbf{v}_k \in P_0 \cap P_1 \subset H$; however, it is easy to see that $\mathbf{v}_0, \ldots, \mathbf{v}_k$ are $k+1$ affinely independent vectors in $\mathbb{R}^k$, and therefore do not all lie in the same hyperplane. This is a contradiction.

We have proved that $\mathbf{v}_k$ lies on a single polytope $P_0$, therefore it can either lie in the interior of $P_0$ or on the boundary of $[0,1]^k$; however, by the construction of each $\epsilon_i$, and by the triangle inequality, it is easy to see that $\mathbf{v}_k \in B_{\ell_1}(\mathbf{v}_0, \epsilon_1) \subset (0,1)^k = \mathrm{int}([0,1]^k)$. We conclude that $\mathbf{v}_k \in \mathrm{int}(P_0)$. $\square$

Let $\delta = \min\limits_{1 < i \leq k} \left( \dfrac{\epsilon_i}{\epsilon_{i-1}} \right) < 1$.

**Proposition A.4.** *The interior of $P_0$ contains all points of the form $(x + y_1, \ldots, x + y_k)$ for which:*

  *(a) All $y_i$ are positive, and are smaller than $\frac{\epsilon_1}{2}$.*

  *(b) The ratio between $y_{i+1}$ and $y_i$ is smaller than or equal to $\delta$.*

*Moreover, each such point is in $\mathbb{R}^k_{sorted}$.*

*Proof.* By Proposition A.2, $\mathbf{v}_0, \ldots, \mathbf{v}_k \in P_0$. We will show that $(x + y_1, \ldots, x + y_k)$ is a convex combination of the points $v_0, \ldots, v_k$ by finding appropriate coefficients.

Let

$$\alpha_0 = 1 - \frac{2y_1}{\epsilon_1}$$
$$\alpha_i = \frac{2y_i}{\epsilon_i} - \frac{2y_{i+1}}{\epsilon_{i+1}} \quad \text{for} \quad 1 \leq i < k$$
$$\alpha_k = \frac{2y_k}{\epsilon_k}.$$

First, we show that the sum of these coefficients equals 1:

$$\sum_{i=0}^{k} \alpha_i = \left( 1 - \frac{2y_1}{\epsilon_1} \right) + \sum_{i=1}^{k-1} \left( \frac{2y_i}{\epsilon_i} - \frac{2y_{i+1}}{\epsilon_{i+1}} \right) + \frac{2y_k}{\epsilon_k}.$$

Notice that the terms in the summation telescope, as each $-\frac{2y_{i+1}}{\epsilon_{i+1}}$ cancels with the corresponding $\frac{2y_i}{\epsilon_i}$ from the next term. After cancellation, we are left with:

$$1 - \frac{2y_1}{\epsilon_1} + \frac{2y_1}{\epsilon_1} - \frac{2y_k}{\epsilon_k} + \frac{2y_k}{\epsilon_k} = 1.$$

Second, we show that all these coefficients are nonnegative. Clearly $\alpha_0, \alpha_k > 0$. For $1 \le i < k$, we have:

$$\frac{\epsilon_{i+1}}{\epsilon_i} \ge \delta \ge \frac{y_{i+1}}{y_i},$$

where the RHS holds due to condition (b) on $y_i, y_{i+1}$, and the LHS holds from the definition of $\delta$. Consequently,

$$\alpha_i = \frac{2y_i}{\epsilon_i} - \frac{2y_{i+1}}{\epsilon_{i+1}} \ge 0.$$

Third, we show that:

$$\sum_{i=0}^{k} \alpha_i v_i = (x + y_1, \ldots, x + y_k).$$

Let's fix any coordinate $1 \le j \le k$. Then we obtain

$$\left\langle \mathbf{e}_j, \sum_{i=0}^{k} \alpha_i \mathbf{v}_i \right\rangle = \sum_{i=0}^{k} \alpha_i \langle \mathbf{e}_j, \mathbf{v}_i \rangle = \sum_{i=0}^{k} \alpha_i x + \sum_{i=j}^{k} \alpha_i \frac{\epsilon_j}{2}$$

$$= x + \sum_{i=j}^{k-1} \left[ \left( \frac{2y_i}{\epsilon_i} - \frac{2y_{i+1}}{\epsilon_{i+1}} \right) \frac{\epsilon_j}{2} \right] + \left( \frac{2y_k}{\epsilon_k} \right) \frac{\epsilon_j}{2}$$

$$= x + y_j.$$

We have proved that $(x + y_1, \ldots, x + y_k)$ is a convex combination of $\mathbf{v}_0, \ldots, \mathbf{v}_k \in P_0$. By Proposition A.3, $\mathbf{v}_k \in \text{int}(P_0)$. Since the coefficient of $\mathbf{v}_k$ in this convex combination is $\alpha_k > 0$, it follows from (Rockafellar, 1970, Theorem 6.1), often referred to as the Accessibility Lemma, that $(x + y_1, \ldots, x + y_k) \in \text{int}(P_0)$.

Finally, the fact that $(x + y_1, \ldots, x + y_k) \in \mathbb{R}^k_{\text{sorted}}$ follows immediately from the fact that all $y_i$ are positive and $\frac{y_{i+1}}{y_i} \le \delta < 1$. $\qquad\square$

Let $\mathbf{w} = (x + y_1, x + y_2, \ldots, x + y_n) \in \mathbb{R}^n$, where $y_i = \frac{\epsilon_1}{4} \delta^{i-1}$. It is straightforward to verify that these $y_i$ satisfy the conditions of proposition A.4, since $y_i < \frac{\epsilon_1}{2}$ and $\frac{y_{i+1}}{y_i} = \delta$. Consider any $k$ ascending indices $r_1 < \cdots < r_k$ from $[n]$. Construct the point $\mathbf{z} = (w_{r_1}, \ldots, w_{r_k}) = (x + y_{r_1}, \ldots, x + y_{r_k}) \in \mathbb{R}^k$. Note that $\delta < 1$ and thus

$$\frac{y_{r_{j+1}}}{y_{r_j}} \le \delta^{r_{j+1} - r_j} \le \delta.$$

This point satisfies the conditions of Proposition A.4, and therefore $\mathbf{z} \in \text{int}(P_0)$. This concludes the proof of Theorem 3.2. $\qquad\square$

## B   A Conjecture for Bounded Multiplicities

The positive result in Theorem 4.3 applies to compact domains of multisets with no repeated points. It is natural to ask whether higher-order Janossy pooling can handle bounded multiplicities. The relevant separation quantity is the distance from a $(k + 1)$-fold multiplicity.

For $A = \{a_1, \ldots, a_n\} \in \mathcal{M}_n(\mathbb{R}^d)$, define

$$r_{k+1}(A) = \min_{\substack{I \subseteq [n] \\ |I| = k+1}} \text{diam}_\infty \{a_i : i \in I\},$$

where

$$\text{diam}_\infty(S) = \max_{u,v \in S} \|u - v\|_\infty.$$

For a domain $D \subseteq \mathcal{M}_n(\mathbb{R}^d)$, set

$$R_{k+1}(D) = \inf_{A \in D} r_{k+1}(A).$$

If $D$ is compact and every point in every multiset $A \in D$ has multiplicity at most $k$, then $R_{k+1}(D) > 0$. The proof is similar to Proposition 4.4. Indeed, if $R_{k+1}(D) = 0$, then there exists a sequence $A_j \in D$ containing $k + 1$ points whose $\ell_\infty$-diameter tends to zero. By compactness, a subsequence of $A_j$ converges in Wasserstein distance to some $A_* \in D$. The $k + 1$ collapsing points converge to the same point in the limit multiset, so $A_*$ contains a point of multiplicity at least $k + 1$, contradicting the assumption on $D$.

We conjecture that this condition is sufficient for $k$-ary CPwL Janossy pooling to be injective, and hence bi-Lipschitz, on $D$.

**Conjecture B.1.** *Let $D \subseteq \mathcal{M}_n(\mathbb{R}^d)$ be compact and suppose that $R_{k+1}(D) > 0$. Then there exist $m \in \mathbb{N}$ and a CPwL function $f : (\mathbb{R}^d)^k \to \mathbb{R}^m$ such that the $k$-ary Janossy pooling of $f$ is injective on $D$.*

Theorem 4.3 proves Conjecture B.1 for $k = 1$. Theorem 3.1 shows that some separation from $(k + 1)$-fold multiplicities is necessary in general: without such a restriction, fixed-order CPwL Janossy pooling is non-injective for every $k < n$. The first unresolved case is $k = 2$, where $R_3(D) > 0$ allows pairs of coincident or near-coincident points but rules out triple multiplicities.

## C  Additional Empirical Details

### C.1  Smooth activation comparison

The main theoretical results in this paper concern CPwL functions. We therefore use ReLU networks in the main experiments. For completeness, we also trained a matched $k = 2$ Janossy autoencoder with GELU activations, using the same architecture and training budget as in Section 5. This comparison is outside the scope of Theorem 3.1, since GELU networks are smooth rather than CPwL, but it is useful for separating two different questions: injectivity and bi-Lipschitz stability.

Prior work shows that smooth multiset functions can have advantages for injectivity, but also face fundamental obstructions to bi-Lipschitzness on multiset domains (Amir et al., 2023; Cahill et al., 2024). This is one of the motivations for focusing on non-smooth CPwL functions in the present paper: when CPwL multiset functions are injective on compact domains, they are automatically bi-Lipschitz under mild conditions, while smooth invariant maps do not generally provide the same stability guarantees.

Empirically, replacing ReLU by GELU did not remove the separation-dependent degradation. The GELU model obtained low-separation MSE $1.612 \cdot 10^{-3}$ on $r(X) \in [0, 0.01]$, and high-separation MSE $1.775 \cdot 10^{-4}$ on $r(X) \in [0.07, 0.09]$, a ratio of 9.08. The corresponding ReLU $k = 2$ model obtained $1.372 \cdot 10^{-3}$ and $1.545 \cdot 10^{-4}$, a ratio of 8.88. Thus, in this controlled finite-capacity experiment, smoothness alone does not eliminate the observed sensitivity to low-separation inputs.

### C.2  Two-dimensional non-injectivity experiment for Janossy maps

We next test the common-affine-region mechanism directly in dimension $d = 2$. We sample four independent random ReLU $k = 2$ Janossy maps of the form

$$E(X) = \frac{1}{n(n-1)} \sum_{i \neq j} f(x_i, x_j), \qquad X = \{x_1, \ldots, x_n\} \subset [0, 1]^2,$$

where $n = 10$, $f : \mathbb{R}^4 \to \mathbb{R}^{16}$ is a randomly initialized one-hidden-layer ReLU MLP of width 64, and the sum is over ordered pairs.

For each random map, we generate starter clouds by sampling one instance of each of seven shapes: grid, ring, blob, two clusters, $L$-shape, triangle, and random cloud. Each shape is normalized to $\ell_\infty$-diameter one, randomly rotated, scaled by one of $\{0.003, 0.006, 0.012, 0.025, 0.045\}$, and placed uniformly in $[0, 1]^2$. We then construct a perturbed cloud

$$X' = \{x_1 + h_1, \ldots, x_n + h_n\}$$

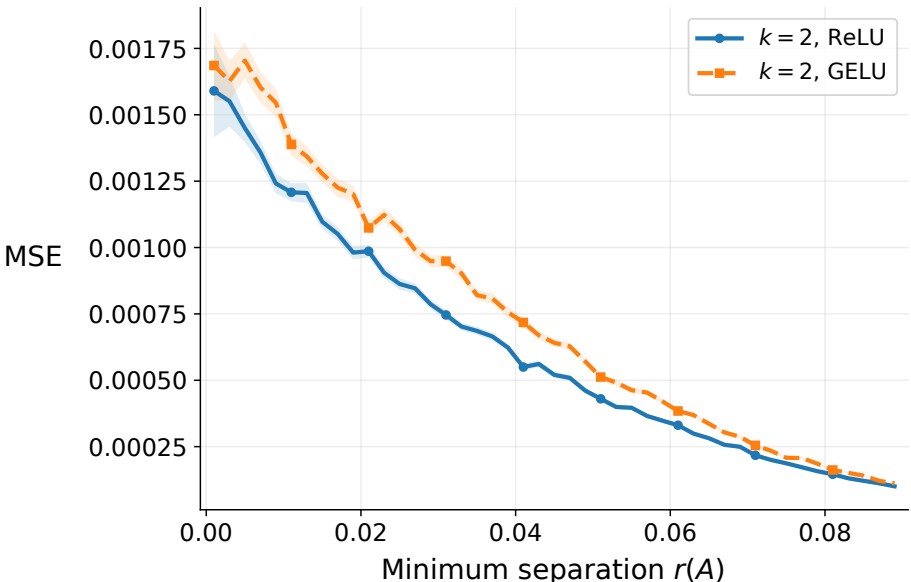

Figure 5: ReLU versus GELU Janossy networks for $k = 2$. The ReLU model is CPwL and is covered by Theorem 3.1; the GELU model is smooth and is included only as an empirical comparison. In this setting, replacing ReLU by GELU does not remove the degradation as the minimum separation approaches zero.

using random perturbations $h_i \in \mathbb{R}^2$ satisfying

$$\sum_{i=1}^{n} h_i = 0.$$

For each cloud we test four perturbation magnitudes, proportional to $0.08, 0.16, 0.30, 0.50$ times the cloud scale. We keep track of whether all ordered pair inputs of $X$ and $X'$ remain in a common ReLU activation region of the map $f$.

Under this common-region condition, such perturbations produce exact equal-output pairs of the Janossy map. Representative examples are shown in Figure 6. In all listed cases, the input clouds are distinct, but their Janossy representations coincide up to floating-point precision.

Across the four random Janossy maps, we generated 560 candidate point clouds, of which 270 satisfied the common-region condition. This high number is expected for random ReLU maps: point clouds with sufficiently small diameter concentrate all ordered pair inputs in a small neighborhood near a point $(p, p)$ on the diagonal, which generally lies in the interior of a linear region. Theorem 3.2 is nontrivial because it must handle arbitrary CPwL region arrangements, such as ones with a boundary on the entire diagonal.

Thus the same common-affine-region mechanism appears in dimension $d = 2$, for independently sampled ReLU Janossy maps and two-dimensional input clouds.

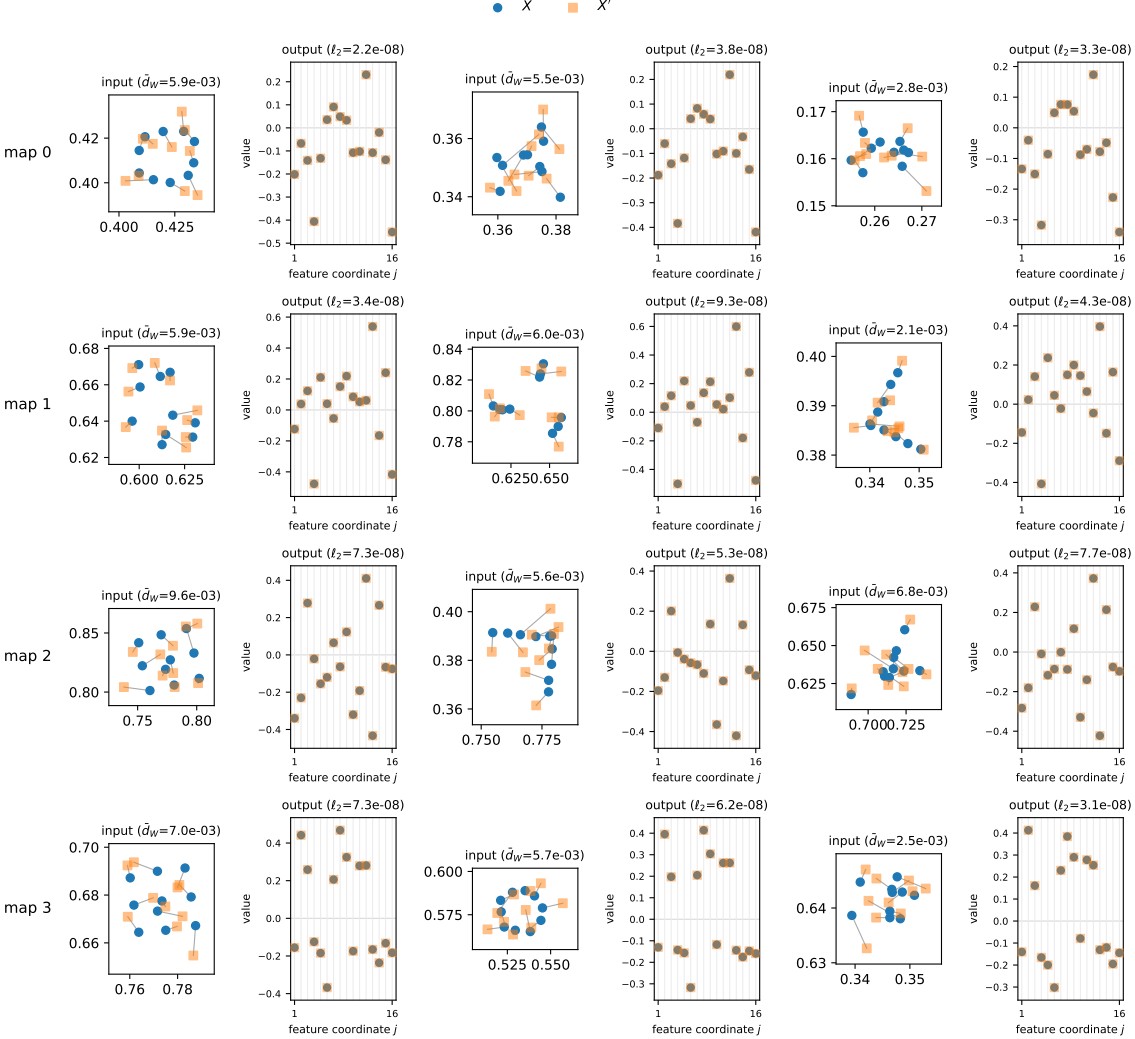

Figure 6: Two-dimensional equal-output pairs for four independent random ReLU $k = 2$ Janossy maps, with one row per random map. In each input panel, blue circles show $X$, transparent orange squares show $X'$, and line segments connect corresponding perturbed points. In each output panel, the horizontal axis is the feature-coordinate index $j$, and the vertical axis is the coordinate value $E_j(X)$ or $E_j(X')$. The two vectors coincide up to floating-point precision.

