# OpenReview forum: "On the (Non) Injectivity of Piecewise Linear Janossy Pooling"
_TMLR — Under review for TMLR_

### Review · Reviewer_zPfu · 2026-04-28

**Summary Of Contributions:**

This paper studies whether continuous piecewise linear (CPwL) k-ary Janossy pooling functions can be injective on multisets. Janossy pooling generalizes both DeepSets (k=1) and Set Transformer-style pairwise models (k=2) by applying a function f to all k-tuples of a multiset and aggregating. The authors prove two main results. First, for any k < n, no CPwL k-ary Janossy pooling can be injective on multisets of size n drawn from a domain containing a line segment: extending the previously known k=1 result to all finite orders. Second, on compact domains where every multiset has n distinct elements with minimum separation R(D) > 0, even 1-ary CPwL Janossy pooling can be made injective and bi-Lipschitz, via an explicit tessellation-plus-local-coordinate construction. The paper closes with a small autoencoder experiment showing reconstruction error grows as the minimum pairwise separation shrinks, and observes that small molecule datasets typically exhibit large enough separation to fall in the "good" regime.

**Audience:**

Yes

**Audience Explanation:**

The paper closes a natural and previously open question in the theory of multiset representations: extending the non-injectivity result of Amir et al. (2023) from k=1 to all k < n. This matters because the bi-Lipschitz multiset embedding literature has been advocating for sorting-based architectures as alternatives to DeepSets and Set Transformers, and a reasonable counterargument was "perhaps higher-order Janossy pooling suffices." This paper rules that path out. Researchers working on injective and bi-Lipschitz GNN architectures now have firm theoretical grounding to pursue sort-based or quantile-based methods rather than searching for clever variants of pairwise pooling. The positive result, while limited in practice by exponential dimension scaling, also clarifies that DeepSets-style pooling is theoretically sufficient in regimes (like small molecules) where points are well-separated, providing post-hoc theoretical justification for an empirically common practice. Theorem 3.2, the combinatorial fact that any polytope partition of $\mathbb{R}^n$ admits a strictly decreasing n-vector all of whose ascending k-subvectors lie in a single polytope interior is itself a non-trivial geometric result that may find independent use.

**Claims And Evidence:**

Yes

**Claims Explanation:**

I worked through the proofs in detail. The core argument in
Theorem~3.2 is the inductive construction of points
$v_0, \ldots, v_k$ via geometrically shrinking $\ell_1$-ball
perturbations
$\bigl(v_i = v_{i-1} + (\varepsilon_i/2)\, e_i$ with
$\varepsilon_i < \varepsilon_{i-1}/2\bigr)$, which yields a sequence
where the polytope memberships nest
$\bigl(\mathrm{POLY}(v_k) \subseteq \cdots \subseteq \mathrm{POLY}(v_0)\bigr)$
and the points are affinely independent. Affine independence in
$\mathbb{R}^k$ forces $v_k$ to lie in the interior of a unique
polytope $P_0$, since $k+1$ affinely independent points cannot
share a hyperplane. The convex combination expressing arbitrary
nearby points in $\mathrm{int}(P_0)$ uses a clean telescoping
coefficient assignment, and the lifting to $\mathbb{R}^n$ relies on
the fact that $\delta < 1$ makes the ratio condition close under
taking subsequences (this last point is implicit in the paper but
mathematically sound).

The reduction from Theorem~3.2 to Theorem 3.1 via symmetrization of
$f$, perturbation analysis on the affine restriction, and a count
of $k$ linear homogeneous equations in $n > k$ unknowns is
straightforward and correct.

The positive result (Theorem 4.3) uses a standard tessellation with
a well-defined CPwL indicator-plus-coordinate construction;
continuity across cell boundaries holds because both sides agree
affinely at shared vertices. The bi-Lipschitz extension via
Sverdlov et al. (2024, Lemma 3.4) is a legitimate application.

I found a small number of expository looseness, Lemma 4.5 should
explicitly state that no element lies in the margin (the proof
establishes this but the statement elides it), and the propagation
of the ratio condition to non-consecutive subvectors could be made
explicit---but no substantive errors. The proofs are sound.

**Requested Changes:**

## Suggested Revisions


1. **Lemma 4.5 statement.** Strengthen the lemma to assert that $F_{Q_k, \mathrm{ind}}(A) = 1$ iff there is a unique $a \in A$ with $a \in Q_k$ *and* no element of $A$ lies in the $\delta$-margin of $Q_k$. The proof already establishes both, but the current statement only mentions the first.

2. **Ratio propagation in Theorem 3.2.** Make explicit that the ratio condition $y_{i+1}/y_i \leq \delta$ propagates to non-consecutive ascending subsequences because $\delta < 1$, so

$$\frac{y_{r_{j+1}}}{y_{r_j}} \leq \delta^{r_{j+1} - r_j} \leq \delta.$$  Currently this is implicit in the final paragraph of the appendix and a careful reader has to fill it in.

3. **Explicit construction of $y_1, \ldots, y_n$.** The proof asserts the existence of a sequence satisfying the conditions of Proposition A.4 but does not exhibit one. Something like $y_i = (\varepsilon_1 / 4) \cdot \delta^{i-1}$, would make the argument concrete.

4. **Lemma A.1 domain.** The lemma states $v \in \mathbb{R}^k$ but writes the ball as $\{x \in [0,1]^k : \|x - v\|_1 < \varepsilon\}$. Either restrict the hypothesis or drop the $[0,1]^k$ from the ball definition for consistency.

5. **Rank of the linear system in Theorem 3.1.** The phrase "$k$ linear homogeneous equations in $n > k$ variables" is correct but understates the structure. Briefly note that since there are fewer equations than unknowns, the kernel is at least $(n-k)$-dimensional regardless of rank, so a nonzero $\boldsymbol{\delta}$ exists.

6. **Wasserstein metric clarification.** Definition 4.1 uses $\ell_\infty$ as ground metric and sums over indices, which is one specific Wasserstein-1 variant. Note this explicitly so readers do not assume the standard Euclidean ground metric.


7. **Stronger empirical validation.** The current experiment shows MSE growing from $\approx 0.0015$ to $\approx 0.0035$ on a 1D toy problem, which is suggestive but not decisive. Two complementary additions would help: (a) construct an explicit pair of distinct multisets that the trained encoder collapses to nearly identical embeddings (using the constructive proof of Theorem 3.1 to produce them), and (b) run on a higher-dimensional task or a real benchmark (e.g., a small graph regression task) where the consequences of non-injectivity should manifest if they manifest anywhere.

8. **Sharpen the gap between non-injectivity and practical impact.** Theorem 3.1 is a worst-case existence statement. Set Transformers nonetheless work well empirically. A discussion of why the theoretical collisions are or are not observed in typical training distributions would help readers calibrate the practical significance.

9. **Address the open conjecture more concretely.** The conjecture that $k$-ary CPwL Janossy pooling is injective on domains where multiplicity is at most $k$ is the natural next theorem and the most practically relevant statement in the paper. Even partial progress (e.g., the case $k=2$, or a counterexample) would strengthen the contribution substantially.

10. **Clarify the cost of Theorem 4.3.** The dimension $m \sim (1/R(D))^d$ is mentioned briefly in Section 4.1 but its severity deserves more emphasis. For $d = 3$ and $R(D) = 0.1$, this is on the order of $10^3$ output features per point.

---

> ### Author Response · Authors · 2026-07-15
>
> We thank the reviewer for the exceptionally careful reading of the proofs and for identifying several places where arguments that were mathematically present could be made fully explicit. We have implemented the requested local proof clarifications and expanded both the empirical validation and the discussion of practical scope.
>
> 1. **Lemma 4.5 statement:** We strengthened the statement of lemma 4.5 according to the suggested revision.
>
> 2. **Ratio propagation in Theorem 3.2:** We added the explicit calculation for the ratio propagation to the text.
>
> 3. **Explicit construction of $y_1, \dots, y_n$:** We adopted the suggested construction.
>
> 4. **Lemma A.1 domain:** We removed the inconsistent restriction.
>
> 5.  **Rank of the linear system in Theorem 3.1:** We now state *"Since there are fewer equations than unknowns, the kernel is at least $(n-k)$-dimensional, so a nonzero solution $\boldsymbol\delta$ exists."*
>
> 6. **Wasserstein metric clarification:** Definition 4.1 now states explicitly that we use the Wasserstein metric with $\ell_\infty$ ground cost. We explain why this choice is natural for the cubical CPwL construction. We also explain why we do not lose generality by using this version of $d_W$ instead of the $\ell_2$ ground cost, thanks to the equivalence of norms:
> $$d_W​(A,B) \leq d_{W,2}​(A,B)\leq \sqrt{d}\cdot
> ​d_W​(A,B)$$
>
> 7. **Stronger empirical validation:** First, in Section 5.2 we constructed example pairs which our trained encoder collapses to nearly equal outputs, using the nullspace perturbations from the proof of Theorem 3.2. We also construct a more separated 'relaxed' example, where we did not strictly require all tuples to fall in the same linear region, but still applied the same nullspace perturbation, and we still observed separation degradation. Second, in Appendix C.2 we give a two dimensional experiment where we randomly initialize four k=2 ReLU Janossy maps, and explicitly find input point clouds which collapse into the same output using the common linear region mechanism. We present a gallery of such inputs.
>
> 8. **Sharpen the gap between non-injectivity and practical impact:** We expanded the conclusion to discuss this point. Three points we mention are: 1. Universal approximation results show that popular models can approximate target functions deliberately well, 2. Many ML tasks do not require injectivity, and 3. Practical data distributions can remain far from repeated-point configurations.
>
> 9. **Address the open conjecture more concretely:** We added Appendix B, where we state the conjecture in a more rigorous manner. We introduce the distance from a (k+1)-fold multiplicity:
>
> $$R_{k+1}(D) = \inf_{A \in D} \min_{I\subseteq [n], \vert I \vert = k+1} \text{diam}_\infty \lbrace a_i : i \in I\rbrace,$$
>
> which is the equivalent to our $R(D)$ from Section for k=2. Compactness together with multiplicity at most $k$ implies $R_{k+1}(D) >0$. We conjecture that this is sufficient for injective CPwL $k$-ary Janossy Pooling.
>
> 10. **Clarify the cost of Theorem 4.3:** We now clearly state the construction dimensions is $\approx (d+1)(2/R(D))^d$. For the QM9 setting, this gives an output dimension of 32000. We clarify in the paper that this is the cost of our specific construction, and that more efficient ones may be possible.
>
> We hope this response has addressed your concerns, and we thank you again for your review.

---

### Review · Reviewer_49cs · 2026-05-11

**Summary Of Contributions:**

The paper proves a negative result on the injectivity of Janossy pooling for multisets, and a positive (constructive) one for sets with near-degenaracy, also discussing how bad things go when the minimal distance between points decreases.

Theory is supported by a (modes) experiment.

**Audience:**

Yes

**Audience Explanation:**

1.

This was initially my main concern, but a quick look at the cited papers in the same family seems to indicate interest in the big conferences (IMCL, ICLR, AAAI).
Yet, the connection to ML problems is not made convincingly/precisely, in particular, if we can have injectivity (and why not bi-lipshitz-ness) for sets (not multisets), then, for most applications, we're happy. I mean in practice, who in ML has multisets (sets with degeneracy)?
However, the scaling of how bad things go (at worst) when the minimal separation between points decreases too much seems very relevant.
In conclusion, it would be very nice to add a paragraph or two (in the conclusion or towards the end of the paper) stating clearly: who in ML may make use of this, what goes wrong in practice when there are near-duplicates, and what the paper implies in those cases. And possibly a few clear take-away message for practitionners.

2.

Relatedly, you could be interested in connecting with architectures such as MACE,
> MACE: Higher Order Equivariant Message Passing Neural Networks for Fast and Accurate Force Fields
Ilyes Batatia, David P Kovacs, Gregor Simm, Christoph Ortner, Gabor Csanyi
Advances in Neural Information Processing Systems 35 (NeurIPS 2022)

which deal with QM9-style datasets and use a form of $\nu$-ary pooling (where $\nu$ is what they call the body order), achieving very nice results.


3.

Related to previous point: I think boosting the experiment section with a few more (I think easy to do) experiments could strenghten a lot the paper and the interest in the community.

**Claims And Evidence:**

Yes

**Claims Explanation:**

Yes, the proofs are sound and overall rather intuitive.

The short experimental section can be easily improved, see suggestions below.

**Requested Changes:**

1. As said above, extend conclusion by relating to applications & take home messages for practionners. Possibly discuss the connection with MACE-style architectures.

2. Please extend the description of the experimental setup, as the task is presently not clearly defined. In particular, what is the ground truth of the labels that are being regressed? Here are some suggestions for improvement:
- define the way the GT targets are designed, explicitly
- define the loss (to clarify how the sorting of the outputs plays a role in reconstruction)
- define how the shaded area is computed
- overlay the theoretical scaling  m ~ (1/R(D))^d (here, d=1, limiting the interest...) over the empirical results (that should then be shown in log-log.)

2. Boosting the experiments section (not just the presentation, but experiments themselves)
- consider extending to larger minimal spearations, to show the trend more clearly. At constant network capacity, I am curous to see the evoltion of the loss at large separation.
- consider extending to k=1 and k>2 (seems trivial to run, from your current codebase - only the input layer of the MLP encoder changes).
- considering at least d=2 dimensionality would be a big plus for the paper. Although I understand this may incurr an important workload, and this is not required, it could bring a lot of added value to the paper, I believe (in that case, showing the scaling of empirical evidence vs expected worst-case scaling  m ~ (1/R(D))^d becomes even more interesting)

3. (related) Importantly, the link between reconstruction error and non-injectivity is asserted, not demonstrated. Please discuss that point. A complementary experiment could be : the same, but with GeLu or other smooth activation function, instead of ReLU ? Since this should help become injective (as fas as I understood).
Or, yet better: discuss how collisions (confusions between different but very similar input sets) occur and why.. this may have been considered by authors, as they say "k-tuples fall into shared linear regions, inducing collisions per Appendix A" (a mysterious sentence)


5. Details:
- "collisions per Appendix A." : the collisions experiment announced (?) in appendix A is not present there, this piece of sentence is very confusing.
- Define acronym WLOG somewhere (without loss of generality)
- typo, p. 9: can be written a a finite union -> "as a"

---

> ### Author Response · Authors · 2026-07-15
>
> We thank the reviewer for pushing us to sharpen the practical interpretation and the empirical presentation. The revised manuscript makes the machine-learning implications more explicit, while also clarifying what the paper does not imply. We also added several experiments aimed directly at the structural mechanism proved in the paper.
>
> **Practical Relevance**
> >As said above, extend conclusion by relating to applications & take home messages for practitoners. Possibly discuss the connection with MACE-style architectures.
>
> We added a discussion on the connection between many body expansions used in MACE and Janossy pooling to page 1 of the paper. We also extended the discussion in the conclusion as follows:
>
> *These conclusions should not be read as predicting that architectures
> such as Deeps Sets or Set Transformers should fail in practice. These models often perform well in downstream tasks, and theoretically are known to approximate all continuous multiset functions. Moreover, versions of these models which use analytic activations rather than CPwL can achieve injectivity guarantees. However, such constructions will not be bi-Lipschitz.*
>
> *Bi-Lipschitz stability was shown to be useful for downstream tasks which require prediction of invariant metrics or in a low parameter regime. Moreover, CPwL and homogenous multiset mappings which are injective are automatically bi-Lipschitz, and it seems that all known bi-Lipschitz constructions are CPwL. This motivated our focus on $k$-ary pooling using CPwL functions: if we could construct  injective multiset functions via $k$ ary pooling over CPwL functions, then we would automatically obtain the bi-Lipschitzness. Unfortunately, our results show that for general domains, such injectivity is not possible, thus justifying the use of previously suggested injective CPwL multiset embeddings, such as sorting based, FSW, or max-filtering. On the positive side, on domains which do not contain near-repeated points, simple CPwL deepsets models suffice for injectivity and bi-Lipschitz guarantees.*
>
> **Empirical section**
>
> The empirical section has been extended to include k=1,2,3 Janossy autoencoders, and explicit constructions of different multisets whose outputs collide in d=1,2 dimensions.
>
> >define the way the GT targets are designed, explicitly
>
> GT Target now defined explicitly as the sorted input multiset.
>
> >define the loss (to clarify how the sorting of the outputs plays a role in reconstruction)
>
> We define the loss as $\mathcal{L}(X)
>     =
>     \frac{1}{10}
>     \sum_{i=1}^{10}
>     \bigl(
>         \hat x_i - x_{(i)}
>     \bigr)^2$
>
> >define how the shaded area is computed
>
> Added explanation - *The shaded region is one standard
> error of the mean within each separation bin.*
>
> >overlay the theoretical scaling m ~ (1/R(D))^d (here, d=1, limiting the interest...) over the empirical results (that should then be shown in log-log.)
>
> We did not add this overlay because it would compare mathematically different quantities. The theoretical relation
> $m \approx R(D)^
> {−d}
> $
> is a sufficient output-dimension estimate for one explicit injective construction. It does not predict a power law for reconstruction MSE. Plotting $R(D)^
> {−d}$
>  against empirical MSE would therefore imply a quantitative law that the theorem does not support. We instead discuss the construction’s dimensional cost separately in Section 4.1.
>
> **Boosting the experiments**
>
> * We added controlled-separation samples to the data. The largest minimal separation now goes up to 0.9, compared to 0.4 we had before. This is close to the largest separation you can have with 10 points on the interval [0,1] which is 1/9. For all k,
> low-separation inputs had roughly 8× larger reconstruction error than high-separation inputs!
> * We extended the Janossy arity in the autoencoder experiment from $k=2$ to $k \in {1,2,3}$.
> * We added a two dimensional experiment in Appendix C.2. where we randomly initialize four k=2 Janossy maps and explicitly construct distinct point clouds in [0,1]^2 whose outputs coincide using the mechanism in the proof of Theorem 3.1.
>
> **Reconstruction error vs non-injectivity and comparing with smooth activations**
>
> * We agree with the logical distinction between reconstruction error and non injectivity. The reconstruction curve is now presented as "separation dependent degradation", not as a proof of non injectivity.
> * Section 5.2 and Appendix C.2 explicitly construct distinct inputs with equal (or nearly-equal) outputs. We discuss how they occur using the mechanism from theorem 3.2. We believe these experiments provide the missing link.
> * We report a matched autoencoder experiment with GELU activation. It exhibits the same separation dependent degradation as ReLU. This is expected, as previous work established that smooth permutation invariant functions are never bi-Lipschitz.
>
> We also defined WLOG and fixed the adressed typo.
>
> We hope this response has addressed your concerns, and we thank you again for your review.

---

### Review · Reviewer_7y1K · 2026-06-12

**Summary Of Contributions:**

The paper studies multiset functions built via k-ary Janossy pooling with continuous piecewise linear (CPwL) component functions, a family that includes DeepSets (k=1) and Set-Transformer-style pooling (k=2). It contributes two main theoretical results. First (Theorem 3.1), it shows that for any k strictly smaller than the multiset cardinality n, CPwL k-ary Janossy pooling can never be injective on general multiset domains, generalizing a previously known k=1 result. The core technical ingredient is Theorem 3.2, a result about polytope partitions of R^k. Second (Theorem 4.3), it shows that on compact domains where all multisets have distinct, well-separated points, even k=1 (DeepSets) pooling can be made injective and bi-Lipschitz, with an explicit construction whose embedding dimension scales with the minimal separation R(D). The paper complements these results with a small multiset-reconstruction experiment using a 2-ary Janossy autoencoder, showing reconstruction error increases as points become closer together.

**Additional Comments:**

Reviewer note: This paper is far from my area of expertise, and I was not able to evaluate the technical correctness of the theoretical results. My assessment is limited to clarity, presentation, and whether the claims appear internally consistent and reasonably supported. I would weight the other reviewers' technical judgments well above my own.

**Audience:**

Yes

**Audience Explanation:**

I believe this is of interest to anyone developing architectures on sets.

**Claims And Evidence:**

Yes

**Claims Explanation:**

The claims are clearly stated and appropriately qualified, and the proofs are laid out in a structured, step-by-step way that reads as internally consistent. The empirical result points in the direction predicted by the theory (error rises as points get closer together), though it is a single small synthetic setting and is presented as an illustration rather than a broad empirical claim. I did not find anything that looked overstated. I want to be candid, however, that this material is far outside my area, so I am not in a position to verify the correctness of the proofs in any depth.

**Requested Changes:**

These are minor presentation issues and do not affect correctness.
1. Fix minor typos (e.g., "Exampe" in Figure 1) and inconsistent notation (e.g., "dw" vs. "d_W").
2. Not every displayed equation is numbered; consider numbering them consistently.
3. Punctuation around displayed equations is inconsistent (e.g., trailing commas/periods); please make this uniform throughout.

---

> ### Author Response · Authors · 2026-07-15
>
> We thank the reviewer for the assessment.
>
> We hereby address the reviewer's presentation-level requests.
>
> > Fix minor typos (e.g., "Exampe" in Figure 1) and inconsistent notation (e.g., "dw" vs. "d_W").
>
> We corrected the Figure 1 typo. We also fixed other typos we found. We standardized $d_W$ as the Wasserstein notation across the paper.
>
> >Not every displayed equation is numbered; consider numbering them consistently.
>
> We have adopted the convention that only numbers equations that are subsequently referenced, and refrained from numbering equations that are used only in local derivations.
>
> >Punctuation around displayed equations is inconsistent (e.g., trailing commas/periods); please make this uniform throughout.
>
> We have now grammatically integrated each equation into the sentence which it is part of, with trailing commas, periods or no punctuation marks accordingly.
>
> We hope this response has addressed your concerns, and we thank you again for your review

---

### Comment · Reviewer_49cs · 2026-07-15
**Discussion**

I invite the authors to address the requested changes.

---

### Author Response · Authors · 2026-07-15

We thank the Area Chair and reviewers for the careful and constructive evaluation. The reviews consistently recognized the soundness and potential interest of the main theoretical contribution, while asking us to make several proof steps explicit, strengthen the empirical validation, and calibrate the practical meaning of the worst-case result. The revised manuscript addresses all three dimensions. Changes from the original document are marked in blue.

We substantially expanded the empirical section. The revised study now compares k=1,2,3 ReLU Janossy autoencoders, fully specifies the target, loss, architecture, data distribution, and uncertainty bands, and shows an approximately eightfold reconstruction gap between low- and high-separation inputs. More importantly, we added proof-guided near-collisions for a trained k=2 encoder and exact equal-output constructions for two-dimensional random ReLU Janossy maps. A matched GELU vs ReLU experiment shows that smoothness alone does not remove the observed low-separation sensitivity.

Furthermore, we expanded the discussion of practical scope. The theorem does not predict that Set Transformers, higher-body-order molecular models, or other successful invariant architectures must fail on typical benchmarks -- they don't. Usually these models are universal approximators, many tasks do not require a globally injective latent representation, and many data distributions avoid the difficult portion of multiset space. The conclusion is instead a worst-case architectural statement: within the CPwL setting, a model seeking uniform injectivity on general multiset domains must look elsewhere than the Janossy pooling family of functions. In this case, sorting-, quantile-, max-filter-, or otherwise explicitly stable CPwL mechanisms are the way to go. The motivation for focusing on injectivity for CPwL multiset mappings is that such mappings are automatically bi-Lipschitz, while mappings using smooth activations cannot be bi-Lipschitz.